**Data Availability Statement:** Data cannot be shared publicly because of identity protection of

# Economic costs of severe seasonal influenza in Colombia, 2017–2019: A multi-center analysis

**Liliana Castillo-Rodríguez**[1,2☯], **Diana Malo-Sánchez**[3‡], **Diana Díaz-Jiménez**[1☯], **Ingrid García-Velásquez**[2‡], **Paola Pulido**[3‡], **Carlos Castañeda-Orjuela**[1]*

**1** Colombian National Health Observatory, Instituto Nacional de Salud, Bogotá, D.C., Colombia, **2** Área Prevención y Control de Enfermedades CDE, OPS/OMS, Bogotá, D.C., Colombia, **3** Dirección de Vigilancia y Análisis del Riesgo en Salud Pública, Instituto Nacional de Salud, Bogotá, D.C., Colombia

☯ These authors contributed equally to this work.
‡ These authors also contributed equally to this work.
* ccastanedao@ins.gov.co

## Abstract

### Objective

To estimate the economic burden of Severe Acute Respiratory Infection (SARI) in lab-confirmed influenza patients from a low-income country setting such as Colombia.

### Methods

A bottom-up costing analysis, from both third payer and social perspectives, was conducted. Direct costs of care were based on the review of 227 clinical records of lab-confirmed influenza inpatients in six facilities from three main Colombian cities. Resources were categorized as: length of stay (LOS), diagnostic and laboratory tests, medications, consultation, procedures, and supplies. A survey was designed to estimate out-of-pocket expenses (OOPE) and indirect costs covered by patients and their families. Cost per patient was estimated with the frequency of use and prices of activities, calculating median and 95% confidence intervals (95% CI) with bootstrapping. Total costs are expressed as the sum of direct medical costs, OOPE and indirect costs in 2018 US dollars.

### Results

The media direct medical cost per SARI lab-confirmed influenza patient was US$ 700 (95% CI US$ 552–809). Diagnostic and laboratory tests correspond to the highest cost per patient (37%). Median OOPE and indirect costs per patient was US$ 147 (95% CI US$ 94–202), with the highest costs for caregiver expenses (27%). Total costs were US$ 848 (95% CI US$ 646–1,011), OOPE and indirect costs corresponded to 17.4% of the total. The median of direct medical costs per patient was three times higher in elderly patients.

### Conclusion

SARI influenza costs impose a high economic burden on patients and their families. The results highlight the importance of strengthening preventive strategies nationwide in the age groups with higher occurrence and incurred health costs.

patients. Data are available from Ethics Committee (contact via secretariactin-cein@ins.gov.co) for researchers who meet the criteria for access to confidential data.

**Funding:** This work was carried out with funds from the Pan American Health Organization PAHO / WHO Regional Office in Colombia and the National Institute of Health, CEMIN Project Code: 2-2017. All funds or sources of support received during this specific study had no role in the study design, data collection and analysis, the decision to publish or the preparation of the manuscript.

**Competing interests:** All authors declare no conflicts of interest.

# 1. Introduction

Influenza is a viral disease with types A and B causing seasonal epidemics [1]. The highest occurrence and mortality rates are observed among high-risk groups, especially children, elderly, and people who have comorbidities, with a higher risk of serious complications such as pneumonia, bronchitis, and sometimes death [2–4]. Globally, annual influenza epidemics cause between 3 and 5 million cases of serious illnesses and 290,000 to 650,000 deaths [1]. In tropical regions, influenza presentation is variable, from annual epidemics coinciding with local rainy season, semi-annual epidemics, or influenza activity throughout the year [5,6]. In the Americas around 85,100 people annually die from influenza (10 deaths per 100,000 person-years) [2].

To reduce the burden of influenza disease, several strategies have been proposed including vaccinating specific population groups such as children under five, adults over 60, pregnant women, and health workers. However, economic considerations as detailed cost analysis are an essential input to effectively guide the formulation of policies for influenza immunization [7]. Decision makers, particularly in lower- and middle-income countries (LMIC), lack economic data to support influenza vaccine policy decisions, therefore information about both direct and indirect cost impacts due to influenza is needed [8].

Influenza causes a significant number of outpatient and hospital visits putting pressure on health care systems [9]. In children it is reflected in school absenteeism and absenteeism from their parents or caregivers with significant economic burden for both health systems and society [8,10]. In high-income countries such as the United States, the direct medical costs for the treatment of influenza were approximately US$ 10.4 billion for 2013, 30% of them due to outpatient visits [7], while in Italy the cost for seasonal influenza epidemics (1999–2008) was estimated between US$ 0.3 and 2.7 billion per year (annual average of US$ 1.4 billion) [11].

In the Latin American region, seasonal influenza imposes high morbidity and economic burden [2,3]. The meta-analysis performed by Savy and colleagues reported an average direct cost of hospitalization of US$ 575 per lab-confirmed influenza case [3], but the source of estimation is not published. However, most cost estimates have been made in patients with clinical manifestations related to influenza infection causing Severe Acute Respiratory Infection (SARI) or Acute Respiratory Infection (ARI), without laboratory confirmation [12,13].

In Colombia, influenza circulates the entire year. Between 2014 and 2018, epidemiological peaks have been presented interspersed. In 2015 and 2017 the epidemiological peak occurred in the first weeks of the year with a predominance of influenza AH3 and AH1N1 pdm09, while in 2016 and 2018 the peak was in May and June with a greater number of reported positive cases in 2018. In 2019 the behavior was similar to odd years with a predominance of influenza AH3 [14]. The objective of this analysis was to estimate the economic costs of SARI in Colombia, from both third payer and social perspectives, assessing direct medical and non-medical costs in patients with lab-confirmed influenza.

# 2. Methods

## 2.1. Study design and perspective

A bottom-up costing study was carried out based on the review of the clinical records of lab-confirmed influenza patients who attended reference hospital centers in three main Colombian cities: Bogotá, Cali and Medellín. The presented analysis is a partial economic evaluation [15] from the third payer (Colombian Health System) and societal perspectives including direct medical and non-medical costs with out-of-pocket expenses (OOPE). Bottom-up costing, also known as ingredient-based analysis or micro-costing, is the most detailed technique

for costing that is based in identification of every single activity and service consumed by the patient and could capture local variation in cost [15,16].

## 2.2. Colombian ARI surveillance system

In Colombia, respiratory disease surveillance includes ARI as an event of public health interest under four strategies [17]: national collective of weekly hospitalized and ambulatory ARI morbidity, national individual immediate ARI mortality in children under five years, sentinel individual weekly influenza-like illness (ILI) and SARI, and national individual immediate unusual SARI surveillance. Sentinel surveillance aims to know the viral circulation and is carried out in 12 health facilities in 10 cities prioritized by the Ministry of Health and National Institute of Health based on geographic location, access to general population, installed capacity for virologic confirmation, and criteria for logistical capacity.

For selection of health facilities included in this analysis the availability of Polymerase Chain Reaction (PCR, real-time or conventional) or immunofluorescence assay (IFA) as diagnosis techniques and the number of positive cases notified in the previous year were applied as selection criteria. Three cities' health facilities were selected: Bogotá (Cardioinfantil Foundation and Colombia University Clinic), Cali (Imbanaco Medical Center and Departmental University Hospital), and Medellín (San Vicente Foundation Hospital and Bolivarian Pontifical University Clinic).

## 2.3. Population

The study population was all patients admitted by emergency room or outpatient clinic that met the clinical SARI definition (patient with fever and cough of less than 10 days requiring hospital management). Samples were collected to perform the confirmatory influenza virus test via a positive laboratory result [17] during the 2017–2019 period.

## 2.4. Patients' selection

Patients for the study were selected in two strategies. The first was patients who were lab-confirmed influenza patients during 2017–2019 and treated in selected centers but that at the time of the start of the study had already been discharged (retrospective recruitment). These patients were selected for the estimation of direct medical costs but did not participate in the estimation of OOPE and indirect costs. The second was patients identified from the start of the study by active search (prospective recruitment) in each included health facility with suspected SARI consultation and subsequently hospitalized. This second group participated in both costing analyses, direct and OOPE.

A random sample of at least 50 patients per city from the health institutions participating in the study was estimated. Thus, a total 227 patients were included in the analysis: Bogotá (Cardioinfantil Foundation n = 38 and Colombia University Clinic n = 57); Cali (Imbanaco Medical Center n = 67 and Departmental University Hospital n = 8); and Medellín (San Vicente Foundation Hospital n = 18 and Bolivarian Pontifical University Clinic n = 39). The capture of participant information was carried out during October to December 2018 and April to June 2019.

For all participants, researchers had access to each medical record and extracted the data about the direct medical costs (S1 Tool). In addition, only the prospective enrolled patients were contacted once they were discharged to conduct information collection about the direct non-medical costs (S2 Tool) and after accepting participation and filling out the informed consent. If a patient was under 18 years old, the information was requested from their parents or legal guardian. The survey was arranged in both a physical and electronic formats through

KoBoToolbox [18], a web-enabled form that was piloted and shared with the enumerators in the three cities for the simultaneous registration and tabulation of data. Thirty days after discharge, patients were re-contacted via telephone and information related to additional expenses during this period was collected in the same form (S2 Tool).

Foreign patients, those whose medical history was incomplete, or who had been referred by this event from or to another health facility, as well as those who did not sign the informed consent were excluded of the analysis.

### 2.5. Costs' estimation

**2.5.1. Direct medical costs.** Estimation of costs of care per patient was based on a bottom-up costing approach [15]. An instrument for data collection was developed in Microsoft Excel® (S1 Tool) and underwent a pilot testing for validation. Information about resource use and frequency of consultations, medications, clinical and paraclinical examinations, and procedures or interventions performed during the SARI lab-confirmed influenza hospitalization episode were identified. No information about volunteer time or donations was collected.

*Resource use.* From the clinical records information about sociodemographic, clinical attention (including intensive care unit -ICU- requirement), and resources' frequency of use was extracted according to six cost items: hospital length of stay (LOS), diagnostic and laboratory tests, medications, consultations, procedures, and supplies (S1 Table). Dosage, total amount, route of administration, and presentation were considered for medications.

*Costs.* Medical billing records were collected to establish direct medical costs. The cost of each item was provided by health facilities in each city. When the cost was not provided by health institution, the tariff manual of the Mandatory Traffic Accident Insurance (SOAT acronym in Spanish) 2018 [19] was consulted. To estimate the medications' cost, the recommendations of the Institute for Health Technology Assessment (IETS acronym in Spanish) were followed, identifying the active principle and the Unique Drug Code from the list of the National Institute of Drug and Food Surveillance (INVIMA in Spanish). It was cross checked with the database of the Drug Price Information System (SISMED in Spanish) to obtain the sale price. The weighted average was estimated by number of units reported [20]. Oseltamivir® cost was reported by the Ministry of Health [21].

**2.5.2. OOPE and indirect costs.** Estimation of direct non-medical costs was based on World Health Organization (WHO) recommendations [22]. A survey was carried out asking general information, pregnancy status of women, number of people in the household, who contributes economically, and monthly income range of patient and family. Information about OOPE before hospitalization included number of days of symptoms before hospitalized, lost days of study or work, time and money spent to reach the health facility, receipt of another type of care and expenses, or if he had to pay a caregiver (S2 Tool). During the hospitalization, the patient was asked about the LOS and any payment done, number of times and cost of the caregiver's displacement, cost of caregiving, and whether this payment affected household finances. Thirty days after hospital discharge, the patient or their relative was re-contacted to know the OOPE after hospitalization including the cost of transportation upon leaving the health institution, as well as if they were re-consulted for influenza, the type of care they received, and expenses for medications, diagnostic and laboratory tests, the cost of the consultation, transportation, and other expenses.

### 2.6. Data analysis

Frequencies of resource use were described using averages, medians, or percentages according to the nature of the variables collected. Characteristics measured on nominal or ordinal scales

were described using proportions. To estimate 95% confidence intervals (95% CI) of the cost results (median cost per patient), Bootstrapping re-sampling techniques were implemented with 50,000 iterations to approximate the actual values of the Gamma distribution of these parameters [23,24]. The mean cost of the resources used was estimated by item and the average cost per patient was calculated as follows:

$$Average\ cost\ per\ patient = \frac{\sum(cost_A) * (frequency\ of\ use_A)}{(n)}$$

where: A = each resource and n = total number of patients

The median and confidence intervals of the cost were estimated by patient, item, age groups, and complexity level of hospitalization (ICU vs. no ICU requirement). All costs were expressed in 2018 US dollars, with an exchange rate of 3,249.75 COP per 1 USD [25]. All data was collected in databases in Microsoft Excel ® and data analysis was performed in R software version 3.6.1. The reported results followed a published Costing Reporting Checklist [26] and validation is presented as supplemental material (S1 Text).

## 2.7. Ethical statement

This research was approved by the Research Ethics Committee of the Instituto Nacional de Salud (CEMIN code 2–2017). According to Colombian Resolution 8430 of 1993 this is a risk-free investigation [27]. The access to the information of clinical records was made as part of mandatory influenza epidemiological surveillance and then by legal mandate, the patients' consent was not required for the use of their information in the analysis [17], although this was obtained for the prospective enrollment. Approval for access to entire clinical records in each health facility was obtained from the health institutions' research/ethical committees. For the application of additional surveys (i.e., OOPE), the physical signature of the written informed consent was requested by the patient or legal representative for children explaining the goal of the research project. In case of illiteracy in a subject, a witness was needed. The records obtained in each health facility were anonymized and codified for the analysis then were never identified in the results with personal or contact information.

## 3. Results

### 3.1. Population characteristics

During the analysis period, a total of 227 lab-confirmed influenza patients in three Colombian cities and six health institutions were identified. Most patients were male, children under five, living in urban areas, affiliated to the contributive health regime, and entered the health institutions with a diagnosis of 'Other acute respiratory lower tract infections' followed by influenza pneumonia. Table 1 shows the main demographic characteristics of the included population. Only three patients corresponded to pregnant women in the cities of Medellín (Bolivarian Pontifical University Clinic) and Cali (Imbanaco Medical Center). The average LOS per patient was 7 days (95% CI 4–12 days) and 24.2% of patients reported comorbidities (Tables 1 and S2).

### 3.2. Direct medical costs

The median cost per patient was US$ 700 (95% CI US$ 552–809). The highest costs were estimated in women, the age group over 65, those affiliated to the special regime, and those who attended the health facilities in Medellín (Table 2). The proportion of patients requiring ICU admission was 12.3% (n = 28) with a cost per patient 12 times higher than who did not require

**Table 1. Characteristics of the population included in influenza SARI analysis in Colombia.**

| Variable | Total | |
|---|---|---|
| **Sex** | n | % |
| Female | 107 | 47.1 |
| Male | 120 | 52.9 |
| **Age group** | | |
| <1 year | 59 | 26.0 |
| 1 to 4 years | 74 | 32.6 |
| 5 to 64 years | 61 | 26.9 |
| > 65 years | 33 | 14.5 |
| **CFR** | | |
| Deaths | 9 | 4.0 |
| **Area** | | |
| Rural | 9 | 4.0 |
| Urban | 218 | 96.0 |
| **Health affiliation regime** | | |
| Contributive | 209 | 92.1 |
| Subsidized | 15 | 6.6 |
| Special | 1 | 0.4 |
| Poor non-insured population | 2 | 0.9 |
| **Admission diagnosis** | | |
| Other | 50 | 22.0 |
| Other acute respiratory lower tract infections | 46 | 20.3 |
| Influenza pneumonia | 44 | 19.4 |
| General signs and symptoms | 42 | 18.5 |
| Acute respiratory upper tract infections | 19 | 8.4 |
| Other diseases of the respiratory system | 15 | 6.6 |
| Chronic diseases of respiratory lower tract | 11 | 4.8 |
| **Comorbidities** | 55 | 24.2 |

ICU admission. By cost item, diagnostic and laboratory tests were the item that generates the highest median cost per patient (37%), followed by LOS (29%) (Table 3).

### 3.3. Direct non-medical costs

The estimation of direct non-medical costs was made from 21 interviews, 52.3% of them were women and 80% belong to contributive regime. The median age of patient was 3.5 years, however, from two different groups <5 and >50 years. Most patients (59.1%) had not yet started school and have neither an occupation nor income. For 36.4% of the patients family income was between 1 and 2 Legal Minimum Monthly Wages (the minimum amount of payment for a formal worker in Colombia). For 77% of patients there were less than five members in the home, and about 30% of those who contribute for the monthly income are both mother and father (S3 Table).

The median OOPE and indirect costs per patient was US$ 147 (CI 95% US$ 94–202) and highest expenses were incurred during hospitalization (US$ 108; CI 95% US$ 76–173). By item, the expenses related to caregiver costs generated the largest proportion (Table 4). No information about the loss of productivity was provided by those interviewed. There was no difference when comparing the median OOPE between children (US$ 149 [CI 95% US $ 93–209]) and adults (US$ 147 [CI 95% US$ 58–219]).

**Table 2. Median direct medical cost per influenza SARI patients in Colombia, according to different disaggregation.**

|  | Median of cost in US$ |
|---|---|
| **All samples** | 700 (CI 95% 552–809) |
| **Sex** | |
| Female (n = 107) | 747 (CI 95% 545–946) |
| Male (n = 120) | 659 (CI 95% 482–809) |
| **Age group** | |
| <1 year (n = 59) | 522 (CI 95% 421–696) |
| 1 to 4 years (n = 74) | 520 (CI 95% 377–720) |
| 5 to 64 years (n = 61) | 758 (CI 95% 437–1,151) |
| > 65 years (n = 33) | 2,319 (CI 95% 1,453–3,472) |
| **Health affiliation regime** | |
| Contributive (n = 209) | 567 (CI 95% 436–720) |
| Subsidized (n = 15) | 522 (CI 95% 286–905) |
| Special (n = 1) | 820 (CI 95% 42–1,596) |
| Poor population non-insured (n = 2) | 447 (CI 95% 23–872) |
| **ICU requirement** | |
| Yes (n = 28) | 7,098 (CI 95% 5,414–9,307) |
| No (n = 199) | 569 (CI 95% 453–696) |
| **City** | |
| Bogotá (n = 95) | 630 (CI 95% 499–809) |
| Cali (n = 75) | 374 (CI 95% 327–439) |
| Medellín (n = 57) | 1,443 (CI 95% 1,163–2,155) |

## 3.4. Total economic cost

The sum of direct medical and non-medical costs was estimated at US$ 847 (CI 95% US$ 646–1,011) per SARI lab-confirmed influenza patient in Colombia from a societal perspective. The OOPE and indirect cost corresponds to 17.4% of the total cost of the disease.

# 4. Discussion

Estimated influenza Severe Acute Respiratory Infections (SARI) costs are important in lower- and middle-income countries (LMIC). However, there are few cost analyses with lab-confirmed patients in hospital settings and both methods and quality are very heterogenous across analyses [28]. Our estimation in a Colombian population account for a total cost per SARI lab-confirmed influenza hospitalized patient of US$ 847 (CI 95% US$ 646–1,011). This bottom-up costing analysis was carried out from direct data of clinical records in six health facilities from

**Table 3. Direct medical costs per patient and item for influenza SARI in Colombia.**

| Item | % | USD $ |
|---|---|---|
| Diagnostic and laboratory tests | 37 | 256 |
| Length of stay (LOS) | 29 | 206 |
| Consultation | 14 | 98 |
| Medicines | 10 | 69 |
| Supplies | 8 | 53 |
| Procedures | 3 | 18 |
| **Total** | **100** | **700** |

**Table 4. Out-of-pocket expenses (OOPE) and indirect costs per item for influenza SARI patient in Colombia.**

| Category | % | US$ |
|---|---|---|
| Caregiver expenses | 27 | 40 |
| Transport | 25 | 37 |
| Other | 16 | 24 |
| Medicines | 11 | 16 |
| Co-payment | 10 | 15 |
| Supplies | 7 | 10 |
| Consults | 4 | 6 |
| Diagnostic and laboratory tests | 0 | - |
| **Total** | **100** | **147** |

three main Colombian cities. This value expressed in 2018 international dollars (using an exchange rate of COP$ 1312.74 per I$) [29] corresponds to I$ 2,099 (95% CI I$ 1,598–2,503). The out-of-pocket expenses (OOPE) corresponds to 17.4% of the total cost of the disease (I$ 365; 95% CI I$ 232–499). No information about the loss productivity was reported by the interviewed patients.

To our knowledge, this is the first multi-center costing study of SARI lab-confirmed influenza cases in the Colombian general population. Some estimations have been performed for influenza vaccine cost-effectiveness analysis in extreme ages (< 2 years and > 65 years) from bottom-up costing in all-cause pneumonia inpatients, but without information reported of cost per case [30]. More recently, a SARI cost analysis in pediatric patients diagnosed by PCR and viral culture from a hospital in Cartagena reported a total direct medical cost (excluding OOPE and indirect costs) greater than us by 45% (I$ 2,523; IQR I$ 1,228–4,810), however the OOPE was 75% higher in our estimation, compared with the I$ 85 (IQR I$ 55–120) of Salcedo et al. They also included a loss productivity evaluation ($ 152; IQR I$ 76–247) [31]. Differences between results could be explained by heterogeneity in sample sizes (44 vs 227 patients included), age structure (only children in Cartagena), and inclusion of contributive regimen population in our estimation.

For the sake of comparability, it is important to consider the difference in influenza cost estimates which may reflect country-specific characteristics, study designs, case identification strategy, study population (in- or out-patients and age groups), and types of costs included in the analysis (direct or indirect) [9]. Our total cost per in-patient is lower than estimates in China and Spain, but higher than Bangladesh and Kenya. China estimated a mean total direct medical cost of SARI hospitalization of I$ 4,172 (I$ 186–63,952) [32] and I$ 21,509 (95% CI, I$ 14,584–31,728) for influenza AH7N9 [33]. In Spain, inpatients with confirmed AH1N1 influenza reported a cost of I$ 8,115, while Bangladesh reported I$ 222 per hospitalization [34] and Kenya I$ 170 [35]. In Shapovalova et al., a metanalysis of 34 studies reported inpatient cost of 2012 I$ 750–9043 (with an outlier for China I$ 113–45,840) [28].

According to age groups, our cost reported in children under one year old was I$ 1,293 (95% CI I$ 1,042–1,723) and for 1–4 years old I$ 1,111 (95% CI I$ 863–1,650), both higher than estimated for these age groups in China (I$ 536) [32], but below that in Spain during the 2009 AH1N1pandemic (I$ 8,702, SD I$ 6,919) [36], and US (I$ 6,583 including I$ 1,704 of productivity loss (39). In the other hand, Zhou et al, found that the cost figures in adults and elderly were I$ 1,983 (IQR I$ 2,066) and I$ 5,254 (IQR 18,116), respectively [32], similar to our findings (in adults I$ 1,877, 95% CI I$ 1,082–2,849 and in elderly I$ 5,741, 95% CI I$ 3,598–8,595). In adults and elderly groups, higher costs were reported in Spanish AH1N1 patients: I$ 11,539 (SD I$ 10,491) for 17–64 years and I$ 13,826 (SD I$ 19,010) in > 65 years [36]. In

general, studies conducted by other authors reported that the estimated costs are higher in the elderly, similar to our results [32,36,37]. For example, our higher costs reported in Medellin is related with an older population in the sample from this city.

Cost estimations and differences across analyses are associated with some cost drivers as length of stay (LOS), comorbidities, and proportion of inpatients requiring Intensive Care Unit (ICU). The overall median LOS per patient in our study was 7 days (95% CI 4–12 days), similar to that reported in China (6 days) [32], but lower than that estimated in Colombian children (9 days; 95% CI, 6.3–11.5) [31]. The hospital LOS was greater in the elderly (7.5 days), but this estimation was less than other research (14 [32] and 17 [4] days). In our analysis, 12.3% of the total patients required ICU showing higher costs than patients who were hospitalized in a general ward (I$ 17,571; 95% CI I$ 13,402–23,039 for ICU patients). As for comorbidity, 24.2% of our patients reported some type of disease while in Spain the percentage of comorbidity was approximately three times higher than ours (64%) [36].

According to the cost item, higher costs were reported for diagnostic and laboratory tests (37%), in contrast to other studies which reported a higher cost for medications (more than 50%) followed by diagnostic and laboratory tests [32,33]. Studies whose population were children, in the US the item with the highest costs comes from LOS and supplies charges 64% [38] as well as previous results in Colombia where the stay corresponds to 30% of the direct medical cost [31]. However, in our study, the hospital LOS was in the second costs item (29%). There are results in the children that report medications with the greatest weight within the direct medical cost with 77% followed by laboratories (37%) [13]. It is likely that the differences are due to those included in the costing in each item, which differs between countries according to the health system and the way in which social security operates [36,39], since in some there are subsidy bonds to specific population groups somehow affecting comparability.

The OOPE are generally hard to measure. In our results it represented 17.4% of the total direct cost, most of them due to caregiver expenses, and incurred during the hospitalization stage. This estimate is greater than what was reported in the Salcedo study, which corresponds to 2.8% of the total direct cost, perhaps due to the fact that in our study all the health regimes and the direct costs of the 227 patients were found, while Salcedo performed the study in subsidized population in addition to having only the direct and indirect costs of 38.6% of all hospitalized patients [31]. In hospitalized children under five years of age it was found that the OOPE corresponded to 4.5% of direct costs with a greater weight in medical care (> 43% of the total OOPE), followed by transportation (> 24%) [38]. Our figures are higher perhaps because our study included additional costs not considered in other studies.

Considering that the family income of the surveyed patients is between 1 and 2 minimum monthly wages, this cost can deeply affect the finances of homes. Strategies to alleviate this expense in families could be studied by reimbursing a percentage of the costs by applying regulated policies for low-income patients. However, full reimbursement in China showed that patients with AH7N9 virus presented higher hospitalization rates after adjusting for disease severity and average admission relative to the other patients [33].

Among our limitations, it is worth to mention: First, the direct medical costs and OOPE of patients with SARI due to influenza were described without being able to estimate the loss of productivity, because the population of surveys was in majority children. Nevertheless, knowing the loss of working time for patients with SARI and their families would present a more complete description of the economic burden of the disease. Second, it was not considered if participating health facilities had an internal protocol that guides the performance of the confirmatory test for respiratory viruses, which probably has skewed the selection of patients with certain characteristics. For example, in Medellín most of the patients belong to an older age group with the presence of comorbidities that can be exacerbated with the presence of the

virus, however disaggregated analyses by different population age groups were carried out. Third, other studies differentiated by type of influenza virus (A or B) and it is likely that results would have been obtained that would further expand the complexity of the study and its findings.

## 5. Conclusions

This study described the direct medical and non-medical costs of care for inpatients who attended medical services with diagnosed SARI lab-confirmed influenza which serves as a tool to know the economic costs of the disease and better identify the likely benefits of potential allocated resources for its prevention. Direct costs covered by health system in Colombia reached 82.6% of total costs, but importantly the economic burden is covered by patients' families. The requirement of ICU admission by patients is a relevant variable as they presented a cost 13 times higher compared to those in the general ward which is most likely due to the combination of comorbidities increasing the severity of the disease.

## Supporting information

**S1 Table. Items included in the direct cost of hospitalized influenza cases.**
(DOCX)

**S2 Table. Comorbidities in the sample of direct costs.**
(DOCX)

**S3 Table. Characteristics people´s surveyed: Indirect costs and out-of-pocket expenses.**
(DOCX)

**S1 Tool. Direct costs survey in the framework of surveillance of ARI in Colombia, 2018.**
(DOCX)

**S2 Tool. Survey of indirect costs and pocket expenses in the framework of ARI surveillance in Colombia.**
(DOCX)

**S1 Text. Costing reporting checklist.**
(DOCX)

## Acknowledgments

The authors would like to give a very special thanks to the persons and institutions that participated in the realization and execution of this project. They were a fundamental piece generating and obtaining of data that made this publication possible. Fabián Guevara and Carolina Villalba Toquica at Clínica Colombia; Rodolfo Dennis Verano and Zenaida Montañez at Fundación Cardioinfantil; Carmen Elisa Ocampo B. at Centro médico Imbanaco; Carlos Echandia at Hospital Universitario del Valle; Fabián Alberto Jaimes Barragán and Sol Beatriz Abad Faciolince at San Vicente Fundación; Víctor Alejandro Acevedo at Pontificia Bolivariana; and Margarita Rosa Giraldo at Secretaria de Salud de Medellín. We also thanks to Dr. Angel Rodríguez, Regional Flu Group Advisor at PAHO in WDC, by edit for proper English language, grammar, punctuation, spelling, and overall style of the manuscript.

## Author Contributions

**Conceptualization:** Liliana Castillo-Rodríguez, Diana Díaz-Jiménez, Ingrid García-Velásquez, Carlos Castañeda-Orjuela.

**Data curation:** Liliana Castillo-Rodríguez, Diana Díaz-Jiménez, Paola Pulido, Carlos Castañeda-Orjuela.

**Formal analysis:** Liliana Castillo-Rodríguez, Diana Díaz-Jiménez, Carlos Castañeda-Orjuela.

**Funding acquisition:** Carlos Castañeda-Orjuela.

**Investigation:** Diana Malo-Sánchez.

**Methodology:** Liliana Castillo-Rodríguez, Diana Díaz-Jiménez, Carlos Castañeda-Orjuela.

**Project administration:** Liliana Castillo-Rodríguez.

**Resources:** Ingrid García-Velásquez.

**Supervision:** Carlos Castañeda-Orjuela.

**Validation:** Diana Díaz-Jiménez, Ingrid García-Velásquez, Carlos Castañeda-Orjuela.

**Writing – original draft:** Liliana Castillo-Rodríguez, Diana Díaz-Jiménez, Carlos Castañeda-Orjuela.

**Writing – review & editing:** Liliana Castillo-Rodríguez, Diana Malo-Sánchez, Diana Díaz-Jiménez, Ingrid García-Velásquez, Paola Pulido, Carlos Castañeda-Orjuela.

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
