## [Decision Letter · Decision Letter 0]

28 Sep 2021

PONE-D-21-15453Economic costs of severe seasonal influenza in Colombia, 2017-2019: A multi-center analysisPLOS ONE

Dear Dr. Castañeda-Orjuela,

Thank you for submitting your manuscript to PLOS ONE. After careful consideration, we feel that it has merit but does not fully meet PLOS ONE’s publication criteria as it currently stands. Therefore, we invite you to submit a revised version of the manuscript that addresses the points raised during the review process.

The reviewer has requested some additions and revisions, in addition to the items raised by the reviewer, please address the following points before more consideration:

In order to provide a more complete information to our readers on the topic, you can compare Median of cost in US$ between participants characteristics (sex, age groups, ...). 

Use a appropriate reporting guideline such as: "Vaughan K, Ozaltin A, Moi F, Kou Griffiths U, Mallow M, Brenzel L. Reporting gaps in immunization costing studies: Recommendations for improving the practice. Vaccine X. 2020;5:100069." to appraise your reporting style and attach it as supplement file. (https://www.equator-network.org/)

We look forward to receiving your revised manuscript.

Kind regards,

Kamal Gholipour, PhD

Academic Editor

PLOS ONE

Journal Requirements:

2. Please provide additional details regarding participant consent. In the ethics statement in the Methods and online submission information, please ensure that you have specified  what type you obtained (for instance, written or verbal, and if verbal, how it was documented and witnessed)."

3. We note that your study included both a prospective and retrospective component. Please ensure you have provided the source of all the medical records obtained in your study (hospital/database name). In addition, in your ethics statement in the manuscript and in the online submission form, please ensure that you have discussed whether all retrospective data/samples were fully anonymized before you accessed them and/or whether the IRB or ethics committee waived the requirement for informed consent for the inclusion of retrospective patient data. If patients provided informed written consent to have data/samples from their medical records used in research, please include this information."

7. We suggest you thoroughly copyedit your manuscript for language usage, spelling, and grammar. If you do not know anyone who can help you do this, you may wish to consider employing a professional scientific editing service.

Reviewers' comments:

Reviewer's Responses to Questions

**Comments to the Author**

1. Is the manuscript technically sound, and do the data support the conclusions?

Reviewer #1: Yes

Reviewer #2: Yes

2. Has the statistical analysis been performed appropriately and rigorously? 

Reviewer #1: Yes

Reviewer #2: Yes

3. Have the authors made all data underlying the findings in their manuscript fully available?

Reviewer #1: Yes

Reviewer #2: No

4. Is the manuscript presented in an intelligible fashion and written in standard English?

Reviewer #1: No

Reviewer #2: Yes

5. Review Comments to the Author

Reviewer #1: Overall Comments: This original research paper provides an important analysis of the costs of influenza in Colombia. The methods and analysis are sounds; however, the writing needs to be refined throughout to eliminate several grammatical errors.

Introduction

The introduction provides solid background on influenza and the motivation for this analysis. I do not think the paragraph describing specific influenza-related costs in China is relevant and recommend removing this paragraph. See note in Discussion.

Methods

Some readers may not be familiar with the ‘bottom-up’ costing approach used in this analysis. I recommend adding a brief explanation of this term and the advantages of this approach after it is initially introduced in the Methods section.

Why were patients who were selected in the first strategy and estimation of direct medical costs excluded from the estimation of OOPE and indirect costs? Pleas add brief explanation to the text.

Please provide a brief definition of ‘ingredient-based approach’.

Results

The results are well-presented. Do you have information on the types of comorbidities identified? That would be good information to include in the presentation of results.

Discussion

The China data are referenced in the Discussion, so it adds some context to why they were included in the Introduction. I think a sentence or two in the Introduction that tells the reader why it is interesting/valuable to compare costs of influenza across geographies is important would tie the Introduction and Discussion together more clearly.

Please include a brief discussion on why the costs in Medellin might be so much higher than the other hospitals included in the study? Are the patients different? Are the charges different?

Reviewer #2: The paper addresses an interesting issues. While the paper is well written, the paper can make more impacts in the literature if the following issues are addressed:

a) I am still very convinced why we need to know the cost burdens. The intro part can add more justification why this type of study was needed. Only getting funding for a research and carrying it out does not necessarily justify academic importance. I am sure authors can do better job here. I am struggling to find the novelty of this work.

b) While study included some CI to capture the variations, I think some regional variations and the reasons for those variation can be better analyzed.

c) It appears that the view-point of analysis was social, it would be better if the authors explicitly mention that along with why that perspective has been adopted.

d) The literature review part appears to be mostly descriptive, not critically reviewed.

e) Some explanations on why only the patients who are visiting the facility are taken will be helpful.

f) Otherwise, the paper is simple and clear.

6. PLOS authors have the option to publish the peer review history of their article (what does this mean?). If published, this will include your full peer review and any attached files.

Reviewer #1: No

Reviewer #2: No

---

## [Author Response · Author response to Decision Letter 0]

9 Dec 2021

Dear

Dr Kamal Gholipour, PhD

Academic Editor

PLOS ONE

We are providing detailed responses to each comment from the editorial team and reviewers in the following tables. We want to thank you revision and recommendations that make possible a strengthen version of our manuscript. We hope that this new version of the manuscript would be considered for publication in your very respectful journal.

Editorial comments

Journal Requirements:

https://journals.plos.org/plosone/s/file?id=wjVg/PLOSOne_formatting_sample_main_body.pdfandhttps://journals.plos.org/plosone/s/file?id=ba62/PLOSOne_formatting_sample_title_authors_affiliations.pdf.

R/ Adjustment made in the tittle page

2. Please provide additional details regarding participant consent. In the ethics statement in the Methods and online submission information, please ensure that you have specified what type you obtained (for instance, written or verbal, and if verbal, how it was documented and witnessed)." 

R/ Clarification was made in the ethical statement

3. We note that your study included both a prospective and retrospective component. Please ensure you have provided the source of all the medical records obtained in your study (hospital/database name). In addition, in your ethics statement in the manuscript and in the online submission form, please ensure that you have discussed whether all retrospective data/samples were fully anonymized before you accessed them and/or whether the IRB or ethics committee waived the requirement for informed consent for the inclusion of retrospective patient data. If patients provided informed written consent to have data/samples from their medical records used in research, please include this information." 

R/ We declared all the hospitals participating in the analysis in methods, section 2,2: “health facilities were selected as a convenience: Bogotá (Cardioinfantil Foundation and Colombia University Clinic), Cali (Imbanaco Medical Center and Departmental University Hospital), and Medellín (San Vicente Foundation Hospital and Bolivarian Pontifical University Clinic).”

In addition, we provided the code of the approval for the realization of project in the ethical statement, now clarifying that access to clinical records was made through the mandatory epidemiological surveillance: 

“2.7. Ethical statement

This research was approved by the Research Ethics Committee of the Instituto Nacional de Salud (CEMIN code 2-2017). According to Colombian Resolution 8430 of 1993 this is a risk-free investigation (30). The access to the information of clinical records was made as part of mandatory influenza epidemiological surveillance, then by legal mandate, it was not required the patients’ consent for the use of their information in the analysis (18), although this was obtained for the prospective enrollment. Approval for access to entire clinical records in each health facility was obtained from the health institutions’ research/ethical committees. For the application of additional surveys (i.e., OOPE) the physical signature of the written informed consent was requested by the patient or legal representative for children, explaining the goal of the research project. In case of illiteracy subject a witness was needed. The records obtained in each health facility were anonymized and codified for the analysis, them were never identified in the results with personal or contact information.

R/ Now the information match in both sections

R/ The adjustments were done to include all the information in the manuscript or supplemental material

R/ Adjustment made

7. We suggest you thoroughly copyedit your manuscript for language usage, spelling, and grammar. If you do not know anyone who can help you do this, you may wish to consider employing a professional scientific editing service.

Whilst you may use any professional scientific editing service of your choice, PLOS has partnered with both American Journal Experts (AJE) and Editage to provide discounted services to PLOS authors. Both organizations have experience helping authors meet PLOS guidelines and can provide language editing, translation, manuscript formatting, and figure formatting to ensure your manuscript meets our submission guidelines. To take advantage of our partnership with AJE, visit the AJE website (http://learn.aje.com/plos/) for a 15% discount off AJE services. To take advantage of our partnership with Editage, visit the Editage website (www.editage.com)and enter referral code PLOSEDIT for a 15% discount off Editage services. If the PLOS editorial team finds any language issues in text that either AJE or Editage has edited, the service provider will re-edit the text for free.

R/ Language was reviewed along the manuscript by the authors

Reviewers Comments

The reviewer has requested some additions and revisions, in addition to the items raised by the reviewer, please address the following points before more consideration:

In order to provide a more complete information to our readers on the topic, you can compare Median of cost in US$ between participants characteristics (sex, age groups, ...). The table 2 of the manuscript shows the median cost according to participant characteristics, however a statistical comparison between these categories (for example as medians difference) is not the goal of our analysis neither the sample constructed has these statistical power

Use a appropriate reporting guideline such as: "Vaughan K, Ozaltin A, Moi F, Kou Griffiths U, Mallow M, Brenzel L. Reporting gaps in immunization costing studies: Recommendations for improving the practice. Vaccine X. 2020;5:100069." to appraise your reporting style and attach it as supplement file. (https://www.equator-network.org/)

R/ We applied the recommended instrument and provided filled as S1 Text.

Reviewer's Responses to Questions

Comments to the Author

1. Is the manuscript technically sound, and do the data support the conclusions?

Reviewer #1: Yes

Reviewer #2: Yes 

R/ No response needed

2. Has the statistical analysis been performed appropriately and rigorously? 

Reviewer #1: Yes

Reviewer #2: Yes 

R/ No response nedded

3. Have the authors made all data underlying the findings in their manuscript fully available?

Reviewer #1: Yes

Reviewer #2: No 

R/ The new version of the manuscript includes six supplementary files

4. Is the manuscript presented in an intelligible fashion and written in standard English?

Reviewer #1: No

Reviewer #2: Yes 

R/ The new version of the manuscript was edited in English language

5. Review Comments to the Author

R/ No response needed

Reviewer #1: 

Overall Comments: This original research paper provides an important analysis of the costs of influenza in Colombia. The methods and analysis are sounds; however, the writing needs to be refined throughout to eliminate several grammatical errors. 

R/ The new version of the manuscript was edited in English language

Introduction

The introduction provides solid background on influenza and the motivation for this analysis. I do not think the paragraph describing specific influenza-related costs in China is relevant and recommend removing this paragraph. See note in Discussion. 

R/ China mention removed from the introduction

Methods

Some readers may not be familiar with the ‘bottom-up’ costing approach used in this analysis. I recommend adding a brief explanation of this term and the advantages of this approach after it is initially introduced in the Methods section. A new sentence and two references were include in methods: “Bottom-up costing, also known as ingredient-based analysis or micro-costing, is the most detailed technique for costing that is based in identification en every single activity an services consumed by the patient and could capture local variation in cost (16,17)”

Why were patients who were selected in the first strategy and estimation of direct medical costs excluded from the estimation of OOPE and indirect costs? Pleas add brief explanation to the text. 

R/ Adjustment in the text was done: “This second group participated in both costing analysis; direct and OOPE”

Please provide a brief definition of ‘ingredient-based approach’. 

R/ The sentence was rewrite and reference adjusted

Results

The results are well-presented. Do you have information on the types of comorbidities identified? That would be good information to include in the presentation of results. 

R/ We include in the new version of manuscript the table of comorbidities as Supplemental table (S2 Table)

Discussion

The China data are referenced in the Discussion, so it adds some context to why they were included in the Introduction. I think a sentence or two in the Introduction that tells the reader why it is interesting/valuable to compare costs of influenza across geographies is important would tie the Introduction and Discussion together more clearly. 

R/ We removed the reference to China estimation in the introduction. Only it is mentioned in the discussion with comparison purposes 

Please include a brief discussion on why the costs in Medellin might be so much higher than the other hospitals included in the study? Are the patients different? Are the charges different? 

R/ It was included a mention in the discussion: “For example, our bigger costs reported in Medellin is related with an older population in the sample from this city”

Reviewer #2: 

The paper addresses an interesting issues. While the paper is well written, the paper can make more impacts in the literature if the following issues are addressed:

a) I am still very convinced why we need to know the cost burdens. The intro part can add more justification why this type of study was needed. Only getting funding for a research and carrying it out does not necessarily justify academic importance. I am sure authors can do better job here. I am struggling to find the novelty of this work. 

R/ A more detailed justification was highlighted in the second paragraph of the introduction: “To reduce the burden of influenza disease, various strategies have been proposed, including vaccination specific population groups such as children under five, adults over 60, pregnant women, and health workers; however, economic considerations, as detailed cost analysis, are an essential input to effectively guide the formulation of policies for influenza immunization (7). Decision makers, particularly in lower- and middle-income countries, lack economic data to support influenza vaccine policy decisions, then information about both direct and indirect cost impacts due to influenza is needed (8)”

b) While study included some CI to capture the variations, I think some regional variations and the reasons for those variation can be better analyzed. 

R/ Thank you for the comment however it is not the focus of our article, and the sample was not designed for these comparisons. However, we include a mention in the discussion. “For example, our bigger costs reported in Medellin is related with an older population in the sample from this city”

c) It appears that the view-point of analysis was social, it would be better if the authors explicitly mention that along with why that perspective has been adopted. 

R/ In Methods, the section 2.1. Study design and perspective stated: “The present analysis is a partial economic evaluation (16) from the third payer (Colombian Health System) and societal perspectives”

d) The literature review part appears to be mostly descriptive, not critically reviewed. 

R/ The review implemented in the discussion is not a goal of the manuscript and only try to identify and report some critical evidence for discuss our results and postulate some reason for the similarities or differences. 

As we mention in the discussion: “difference in influenza cost estimates, which may reflect country-specific characteristics, study designs, case identification strategy, study population (in- or out-patients and age groups), and types of costs included in the analysis (direct or indirect)”

e) Some explanations on why only the patients who are visiting the facility are taken will be helpful. 

R/ We realized the costing only from severe cases, then only inpatient subjects are included in the analysis: “clinical SARI definition (patient with fever and cough less than 10 days of evolution requiring hospital management)”

f) Otherwise, the paper is simple and clear. 

R/ No response needed

6. PLOS authors have the option to publish the peer review history of their article (what does this mean?). If published, this will include your full peer review and any attached files.

Do you want your identity to be public for this peer review? For information about this choice, including consent withdrawal, please see our Privacy Policy.

Reviewer #1: No

Reviewer #2: No 

R/ No response needed

---

## [Decision Letter · Decision Letter 1]

18 Mar 2022

PONE-D-21-15453R1Economic costs of severe seasonal influenza in Colombia, 2017-2019: A multi-center analysisPLOS ONE

Dear Dr. Castañeda-Orjuela,

Thank you for submitting your manuscript to PLOS ONE. After careful consideration, we feel that it has merit but does not fully meet PLOS ONE’s publication criteria as it currently stands. Therefore, we invite you to submit a revised version of the manuscript that addresses the points raised during the review process.

As you will see, the reviewer felt that your manuscript had improved, but had concerns regarding the writing quality of the manuscript. If you are able to improve the clarity and quality of language in your manuscript, with the assistance of a professional copyediting agency, we would be willing to reconsider a resubmission of your work.

We look forward to receiving your revised manuscript.

Kind regards,

Kamal Gholipour, PhD

Academic Editor

PLOS ONE

Journal Requirements:

Reviewers' comments:

Reviewer's Responses to Questions

**Comments to the Author**

1. If the authors have adequately addressed your comments raised in a previous round of review and you feel that this manuscript is now acceptable for publication, you may indicate that here to bypass the “Comments to the Author” section, enter your conflict of interest statement in the “Confidential to Editor” section, and submit your "Accept" recommendation.

Reviewer #1: All comments have been addressed

Reviewer #2: All comments have been addressed

2. Is the manuscript technically sound, and do the data support the conclusions?

Reviewer #1: Yes

Reviewer #2: Yes

3. Has the statistical analysis been performed appropriately and rigorously? 

Reviewer #1: Yes

Reviewer #2: Yes

4. Have the authors made all data underlying the findings in their manuscript fully available?

Reviewer #1: Yes

Reviewer #2: Yes

5. Is the manuscript presented in an intelligible fashion and written in standard English?

Reviewer #1: No

Reviewer #2: Yes

6. Review Comments to the Author

Reviewer #1: The authors have made the appropriate and requested updates to the article content. There are still several grammatical errors that make for difficult reading. In response to the reviewers, the authors indicated they reviewed the manuscript for grammar edits. I do not think their review was sufficient and recommend using a professional service before publication.

Reviewer #2: Thanks for addressing all comments. The manuscript took a better shape. I believe it would a good contribution, if accepted. Good luck with your work!

7. PLOS authors have the option to publish the peer review history of their article (what does this mean?). If published, this will include your full peer review and any attached files.

Reviewer #1: No

Reviewer #2: **Yes: **Shafiun Nahin Shimul

---

## [Author Response · Author response to Decision Letter 1]

4 May 2022

Bogotá, May 2nd, 2022

PONE-D-21-15453R1

Economic costs of severe seasonal influenza in Colombia, 2017-2019: A multi-center analysis

PLOS ONE 

Dear reviewers and editorial team

Thanks to editorial team and reviewers to provide valuable comments to improve the manuscript quality. We addressed the recommendations and are providing response (in blue) to each editor and reviewers’ comment. 

Reviewers' comments: 

Reviewer's Responses to Questions

Comments to the Author

1. If the authors have adequately addressed your comments raised in a previous round of review and you feel that this manuscript is now acceptable for publication, you may indicate that here to bypass the “Comments to the Author” section, enter your conflict of interest statement in the “Confidential to Editor” section, and submit your "Accept" recommendation.

Reviewer #1: All comments have been addressed

Reviewer #2: All comments have been addressed

R/ No response needed

2. Is the manuscript technically sound, and do the data support the conclusions?

Reviewer #1: Yes

Reviewer #2: Yes

R/ No response needed

3. Has the statistical analysis been performed appropriately and rigorously? 

Reviewer #1: Yes

Reviewer #2: Yes

R/ No response needed

4. Have the authors made all data underlying the findings in their manuscript fully available?

Reviewer #1: Yes

Reviewer #2: Yes

R/ No response needed

5. Is the manuscript presented in an intelligible fashion and written in standard English?

Reviewer #1: No

Reviewer #2: Yes

R/ This version of the manuscript was edit for proper English language, grammar, punctuation, spelling, and overall style.

6. Review Comments to the Author

Reviewer #1: The authors have made the appropriate and requested updates to the article content. There are still several grammatical errors that make for difficult reading. In response to the reviewers, the authors indicated they reviewed the manuscript for grammar edits. I do not think their review was sufficient and recommend using a professional service before publication.

Reviewer #2: Thanks for addressing all comments. The manuscript took a better shape. I believe it would a good contribution, if accepted. Good luck with your work!

R/ Thanks to the reviewers for the comments and the appreciation of our adjustments in this version of the manuscript. Now, we receipt help from WDC PAHO office to edit for proper English language, grammar, punctuation, spelling, and overall style of this version of the manuscript.

7. PLOS authors have the option to publish the peer review history of their article (what does this mean?). If published, this will include your full peer review and any attached files.

Do you want your identity to be public for this peer review? For information about this choice, including consent withdrawal, please see our Privacy Policy.

Reviewer #1: No

Reviewer #2: Yes: Shafiun Nahin Shimul

R/ No response needed

Carlos Castañeda-Orjuela

---

## [Editor Report · Decision Letter 2]

6 Jun 2022

Economic costs of severe seasonal influenza in Colombia, 2017-2019: A multi-center analysis

PONE-D-21-15453R2

Dear Dr. Castañeda-Orjuela,

We’re pleased to inform you that your manuscript has been judged scientifically suitable for publication and will be formally accepted for publication once it meets all outstanding technical requirements.

Kind regards,

Kamal Gholipour, PhD

Academic Editor

PLOS ONE
---

## [Editor Report · Acceptance letter]

10 Jun 2022

PONE-D-21-15453R2 

Economic costs of severe seasonal influenza in Colombia, 2017-2019: A multi-center analysis 

Dear Dr. Castañeda-Orjuela:

I'm pleased to inform you that your manuscript has been deemed suitable for publication in PLOS ONE. Congratulations! Your manuscript is now with our production department. 

Kind regards, 

on behalf of

Dr. Kamal Gholipour 

Academic Editor

PLOS ONE